# Glial PAMPering and DAMPening of Adult Hippocampal Neurogenesis

**DOI:** 10.3390/brainsci11101299

**Published:** 2021-09-29

**Authors:** Luke Parkitny, Mirjana Maletic-Savatic

**Affiliations:** Baylor College of Medicine and Jan and Dan Duncan Neurological Research Institute at Texas Children’s Hospital, Houston, TX 77030, USA; maletics@bcm.edu

**Keywords:** neurogenesis, neural stem cells, neuroinflammation, microglia, glia

## Abstract

Adult neurogenesis represents a mature brain’s capacity to integrate newly generated neurons into functional circuits. Impairment of neurogenesis contributes to the pathophysiology of various mood and cognitive disorders such as depression and Alzheimer’s Disease. The hippocampal neurogenic niche hosts neural progenitors, glia, and vasculature, which all respond to intrinsic and environmental cues, helping determine their current state and ultimate fate. In this article we focus on the major immune communication pathways and mechanisms through which glial cells sense, interact with, and modulate the neurogenic niche. We pay particular attention to those related to the sensing of and response to innate immune danger signals. Receptors for danger signals were first discovered as a critical component of the innate immune system response to pathogens but are now also recognized to play a crucial role in modulating non-pathogenic sterile inflammation. In the neurogenic niche, viable, stressed, apoptotic, and dying cells can activate danger responses in neuroimmune cells, resulting in neuroprotection or neurotoxicity. Through these mechanisms glial cells can influence hippocampal stem cell fate, survival, neuronal maturation, and integration. Depending on the context, such responses may be appropriate and on-target, as in the case of learning-associated synaptic pruning, or excessive and off-target, as in neurodegenerative disorders.

## 1. Introduction

Adult mammalian brains preserve the capacity to generate and integrate new functional neurons through a process called adult neurogenesis. Adult neurogenesis underlies critical aspects of normal brain function and its disruption has been linked to neuropathologies such as epilepsy, traumatic brain injury, schizophrenia, autism, depression, and Alzheimer’s Disease [1,2,3,4,5]. Recent evidence suggests that human brains can generate new neurons into very old age, even in individuals with marked cognitive impairment [6,7]. Since these impairments are associated with diminished neurogenesis, its therapeutic enhancement is a prime target for halting or reversing cognitive decline [6,7].

In mammals, adult neurogenesis has been demonstrated in the neurogenic niches of the subventricular zone (SVZ) of the lateral ventricles and the subgranular zone (SGZ) of the hippocampal dentate gyrus [1]. Here, neural stem and progenitor cells (NSCs, NPCs, respectively), are tasked with the generation of all new neurons, astrocytes, and oligodendrocytes. The fates of these stem/progenitor cells are determined by cell-intrinsic programming and signaling from their complex microenvironment. Some of the most important signals stem from glial cell interactions with neurons and NPCs.

In this review, we seek to review what determines whether an encounter with glia results in the generation and integration of a new neuron. We will discuss the communication pathways and effector mechanisms that glia use to influence adult neurogenesis. We will particularly focus on how these cells are guided by danger and damage signals to help guide the fates of hippocampal progenitors and newborn neurons. Danger signaling refers to a communication system between stressed or damaged cells and immune cells that uses signaling molecules to initiate immune and inflammatory destructive responses. In the context of adult neurogenesis, these signals are related to the cell–cell signals that control the physiologic phagocytic pruning of newly proliferated progenitors and the pathophysiological signaling that occurs in a neuroinflammatory microenvironment, such as in aging and neurodegeneration. Thus, danger signaling provides an apt framework to explore how cell–cell and cell–environment interactions can shape neurogenesis and how these mechanisms might be exploited as part of future therapy development.

## 2. The Main Players in Adult Hippocampal Neurogenesis (AHN)

### 2.1. From Stem Cell to Neuron

AHN arises from the dichotomous differentiation potential of multipotent neural stem cells (NSCs) to self-renew and generate mature neurons that incorporate into the local hippocampal functional circuitry [8,9], or glia such as astrocytes (GFAP+S100β+) and oligodendrocytes in a process termed gliogenesis [10,11,12,13]. The pathophysiological potential of gliogenic fate switching is particularly well demonstrated in epilepsy, where rodent models have shown that NSCs can directly transform into reactive astrocytes, contributing to seizure activity and impairing AHN by suppressing the neurogenic lineage [14].

Considering that the generation of a mature neuron is the ideal outcome of adult neurogenesis, we expect the following cascade of events: NSCs (type-1 radial glialike cells; Nestin + Glial fibrillary acidic protein (GFAP)+SOX2+) divide asymmetrically to generate NPCs (type-2 cells; 2a: GFAP-Nestin + Doublecortin (DCX)-SOX2+; 2b: GFAP-Nestin+DCX+SOX2+]) which either undergo apoptosis or transition into type-3 cells (Nestin-DCX+SOX2-; also called neuroblasts) and eventually form post-mitotic granule cells (Neuronal Nuclei (NeuN)+) that mature and incorporate into the hippocampal circuitry [11,12,15,16,17,18,19,20]. The initiation and magnitude of cell proliferation, differentiation, survival, migration, and circuit integration depends on numerous intrinsic and extrinsic factors, including signaling from immunocompetent cells such as the glia.

### 2.2. The Glia

Microglia, astrocytes, and oligodendrocytes comprise the three main glial cell types. While morphologically and functionally different, these cells closely coordinate to maintain structural integrity and homeostasis in the hippocampus by responding to environmental cues that include danger signaling. Although we focus exclusively on these resident glial cells, it must be noted that other immune cells, such as infiltrating macrophages and CD4+ T-cells, modulate the neurogenic environment, especially in neuroinflammatory contexts [21,22,23,24].

#### 2.2.1. Microglia

Microglia are evolutionarily ancient macrophagelike immunocompetent cells of the central nervous system (CNS) [25]. They assist with the physiologic pruning of neuronal circuits and the identification and eradication of damage and infection in pathophysiologic contexts.

In embryogenesis, microglial cells are the first glia to appear in the CNS. In mice, they migrate to the developing brain from the yolk sac by embryonic day 9.5 and start to express the classic macrophage markers CD45, CD11b, ionized calcium-binding adaptor molecule 1 (Iba1), fractalkine receptor (CX3CR1), and F4/80 (the latter is present in rodents but not humans) [26,27]. In addition to these common macrophage markers, rodent and human microglia can be uniquely identified by their expression of transmembrane protein 119 (TMEM119), which helps to differentiate them from infiltrating brain macrophages [28,29]. In addition, microglia express purinergic receptors that modulate phagocytosis and cell activation in response to damage signaling and environmental cues as well as the colony-stimulating factor 1 receptor (CSF1R), critical to microglial proliferation and survival [30,31]. Because of its functional importance, CSF1R can be exploited for in vivo microglial knockout studies by utilizing the CSF1R inhibitor PLX5622 and the CSF1R/stem-cell factor receptor (KIT)/FMS-like tyrosine kinase 3 (FLT3) inhibitor PLX3397. In humans, microglia populate the developing brain in two waves: at 4.5 weeks of gestation and around eight weeks later [32,33]. Just a few days later, a functional blood–brain barrier (BBB) is established and the CNS microglial population becomes effectively isolated from further infiltration by systemic cells [27]. Due to their spatial isolation, the pool of CNS tissue-resident microglia is renewed and repopulated locally without contribution from the systemic immune pools [34]. In the human brain, carbon dating has shown that microglia are relatively long-lived cells, with only around a third being replaced every year [35]. Resident microglia continue to mature through three additional stages toward the adult immune surveillance phenotype that appears during the postnatal period and characterizes the typical microglia in the healthy adult brain [36]. With aging, animal studies have shown that microglia assume a more primed state that is characterized by slower but exaggerated and prolonged inflammatory activation [37]. Imaging and transcriptomic analyses of postmortem isolated human brains have also demonstrated an aged microglial phenotype [38,39,40]. However, these studies have also suggested that human and rodent microglia appear to age differently and further work is needed to better inform the translational interpretation of cross-species data involving specific functional phenotypes [40].

Microglia are functionally and morphologically malleable phagocytic cells, which allows them to actively monitor and respond to their environment. Resident, non-activated (so-called ‘resting’) ramified microglia have important physiological roles such as synaptic pruning and removal of apoptotic cells through phagocytosis [41,42,43]. Phagocytosis is critical for normal AHN where most newborn cells undergo apoptosis and clearance [42]. It is in essence a response to “find me” and “eat me” signaling by cells and is most efficiently carried out by non-activated sentinel and alternatively-activated M2 microglia. Impaired phagocytosis and neurogenesis have been described in rodent models of various pathologies. In a kainic acid mouse model of mesial temporal lobe epilepsy, microglia were shown to express fewer receptors that are important for apoptotic signaling, resulting in reduced phagocytosis and increased pro-inflammatory cytokine expression; the sequela of which are abnormal epileptic neuronal circuit development and function [44]. Aging has also been associated with a reduced microglial phagocytic capacity of products such as amyloid-β [45], which is consistent with a transition toward a more pro-inflammatory phenotype. However, these changes appear to be stimulus and context-specific, as aged in vitro microglia have been found to exhibit an increased phagocytic capacity of neuronal debris [46].

The second key microglial mechanism is the ability to secrete cytokines, which are immunomodulatory proteins with autocrine, paracrine, and endocrine signaling functions. Given appropriate danger signaling, microglia are activated into an M1 pro-inflammatory state associated with a retracted amoeboid morphology and the secretion of reactive oxygen species and cytokines such as interleukin (IL)-1β, tumor necrosis factor (TNF)-α, and IL-6. Under normal circumstances these cells transition to an M2 anti-inflammatory-repair alternative activation state, associated with the secretion of IL-4, IL-10, IL-13, and transforming growth factor (TGF)-β [47]. However, with aging and neurodegenerative disease, microglia can get stuck in the pro-inflammatory phenotype. Of course, these dichotomous activation categories do not explain the rich spectrum of microglial phenotypic states that exist involving myriad combinations of morphology, gene expression, and receptor expression each of which is determined by physiology, pathophysiology, and even anatomical location [48,49,50,51,52,53]. However, our most important take-away message is that pro-inflammatory pressures on and by microglia can suppress AHN, as will be later described.

#### 2.2.2. Astrocytes

Astrocytes, or astroglia, are vastly abundant in the adult mammalian brain. Given their complex roles in supporting brain homeostasis and modulating neuroimmune and neuronal function, it is not surprising that these cells have very heterogeneous phenotypes. During rodent embryonic development, the first astrocytes appear at the end of the neurogenesis-dominant period at E12–18 [54,55]. In the adult rodent brain, most new astrocytes arise through local cell division of existing differentiated astrocytes, while in the SVZ, new astrocytes arise from progenitors that switch from the neurogenic toward gliogenic lineage [55]. Astrocytes play a critical role in the adult brain by providing metabolic support to neurons, modulating cerebral blood flow, blood-brain-barrier (BBB) maintenance, ion homeostasis, and by sensing and modulating synaptic transmission [56,57,58,59,60]. Recent work suggests that astrocytes also play an important role in apoptotic cell clearance and synapse remodeling through complement component 1q (C1q) receptor MEGF10-dependent phagocytosis [61,62]. 

Classically, astrocytes are defined by their expression of the intracellular markers calcium-binding protein S100β and glial fibrillary acidic protein (GFAP), however, these markers are influenced by regional differences, cell activation, and cell phenotype (e.g., GFAP is reduced in protoplasmic astrocytes) [63,64,65]. GFAP is not a unique astrocytic marker, however, as it is also expressed by NSCs in the dentate gyrus (also known as Type I cells or radial glia-like (RGL) cells) [66]. Other astrocyte-expressed markers include aldehyde dehydrogenase 1 family member L1 (ALDH1L1), glutamine synthetase, glutamate transporter 1, aquaporin-4; however, their expression is not stable [67,68]. Unfortunately, the close familial relationship between astrocytes and NSCs/NPCs has so far precluded the identification of unique surface markers for these cells, limiting our ability to identify and characterize them using live-cell approaches such as flow-assisted cell sorting. 

Astrocytes are traditionally dichotomized into fibrous white-matter and protoplasmic gray-matter phenotypes [69], but in reality, a greater diversity exists within and among brain regions, suggesting that these cells actively adapt to environmental demands [70,71,72]. In the presence of danger signaling, astrocytes adopt a pro-inflammatory neurotoxic A1 reactive state [73]. This can be induced by activated microglial secretion of interleukin (IL)-1α, tumor necrosis factor (TNF)-α, and C1q or damaged mitochondria [73,74]. These A1 astrocytes, in turn, act to limit microglial synapse-supportive and phagocytic functions while releasing neurotoxic products [73]. In short, by responding to danger signaling and environmental cues, astrocytes and microglia can create an environment that is not conducive to the survival of newborn cells. In aging, similarly to the microglial phenotypic shift, A1 astrocytes increase in number, and in mice, suppression of the A1 state has shown promise in the treatment of Parkinson’s Disease and Alzheimer’s Disease (AD) [75,76,77]. Conversely, A2 astrocytes, have a largely neuroprotective role, mediated by the release of neurotrophic products [78].

#### 2.2.3. Oligodendrocytes

Oligodendrocytes are the myelinating cells of the CNS and are not typically considered as part of the immunocompetent glial pool. Starting life as oligodendrocyte precursor cells (OPCs) after the developmental gliogenic switch, they migrate into their terminal sites where they mature into myelinating cells [79,80]. OPCs can be identified by their expressed proteins, neuron-glial antigen 2 (NG2), platelet-derived growth factor receptor (PDGFRα), oligodendrocyte transcription factor (Olig) 1 and 2, and doublecortin (DCX), while mature myelinating cells are marked by their expression of myelin proteins such as myelin basic protein (MBP) [81,82,83].

In addition to their main myelination role, oligodendrocytes modulate neuronal function and provide neuronal trophic support by releasing products such as insulin-like growth factor 1 (IGF-1), brain-derived neurotrophic factor (BDNF), nerve growth factor (NGF), and glial cell line-derived neurotrophic factor (GDNF) [84,85,86,87]. They participate in immune signaling, including glial-glial communication, through expressed Pattern Recognition Receptors (PRRs) [Toll-Like Receptors; TLR2 and TLR4], cytokine receptors [for IL-1, IL-4, IL-6, IL-7, IL-10, IL-11, IL-12, IL-18, and interferon (IFNγ)], chemokine receptors [CXCR1 and CXCR2 (receptors for IL-8/CXCL8 and GRO-α/CXCL1), CXCR3 (MIG/CXCL9, IP-10/CXCL10, and ITAC/CXCL11) and CXCR4 (SDF-1/CXCL12)], and CD200 [88,89,90,91,92,93,94,95]. In specific infectious and inflammatory contexts, oligodendrocytes secrete chemokines and cytokines such as CCL2, CCL3, CCL5, CXCL10, IL-6, IL-8, and IL-18 that modulate chemotaxis and activation of macrophages, microglia, and T-cells [91,96,97]. 

Little attention has been directed toward OPCs and oligodendrocytes in AHN research. The limited available data show them to be worthy targets of attention. Firstly, oligodendrocytes have been demonstrated to emerge from the gliogenic pathway, driven by over-expression of the Achaete-Scute Family Homolog 1 (Ascl1; Mash1) transcription factor, a mechanism that may prove to be important in demyelinating diseases [98,99]. Secondly, OPCs can be made to transiently express the immature neuronal marker doublecortin [83,100]. Whether this suggests a neurogenic potential is not known. Lastly, as described above, oligodendrocytes can function as immunocompetent cells and thus modulate the hippocampal neurogenic niche.

## 3. How Danger Signaling Determines Cell Fate in AHN

Pathogen-associated (PAMPs) and danger-associated molecular products (DAMPs) are ligands that are released by cells during infection, tissue damage, and as part of programmed cell death (apoptosis). Danger signals target immunocompetent cells that carry Pattern Recognition Receptors (PRRs). Some danger signals are involved in pro-phagocytic “find me” and “eat me” signaling; once phagocytes are present, PRR activation can stimulate the production and release of various immunomodulatory products. As such, danger signals are critical in rallying glial cell responses to homeostatic and pathophysiological needs.

We will now discuss how danger signaling modulates AHN under normal physiological conditions and then in response to pathological insults such as during neurodegenerative disease. While the distinction is somewhat artificial, as the involved mechanisms are overlapping, we believe that this approach provides a convenient framework to consider how cell–cell communication in the neurogenic niche affects AHN.

### 3.1. The Healthy Hippocampal Environment

The healthy adult hippocampal microenvironment is largely characterized by glial surveillance, neuroplastic remodeling, and phagocytosis. Within these relatively stable conditions, glia are tasked with pruning synapses and eliminating surplus cells that are generated as part of the AHN cascade. It is worth noting that although synaptic pruning in the uninjured brain is normally thought of as a feature of early development, human data suggest that it continues into at least the third decade of life [101].

#### 3.1.1. Find-Me and Eat-Me Signaling

Surveilling microglia make frequent contacts with NSCs/NPCs and neurons, which allows them to identify targets for phagocytosis and inflammatory responses. To attract microglial attention, cells can present “find me” signals. Phagocytosis is then encouraged through the expression of “eat me” signals.

“Find me” signals include released nucleotides such as adenosine triphosphate (ATP) and the chemokine fractalkine (CX3CL1). CX3CL1 is one of the most important “find-me” signaling pathways between microglia and neurons and involves the soluble form of the chemokine fractalkine (CX3CL1) and its receptor CX3CR1. In mouse and human brains, CX3CL1 is constitutively expressed by neurons and astrocytes as a transmembrane molecule that is proteolytically cleavable into a soluble chemoattractant form [102,103,104,105,106,107,108]. Brain CX3CR1 expression is highest in microglia, although it is also expressed by some infiltrating peripheral immune cells [109]. CX3CL1 signaling modulates immune activity, demonstrating broadly neuroprotective effects in the spinal cord and brain [110,111] while promoting microglial phagocytosis of apoptotic cells [112,113]. In other words, brain CX3CL1 signaling promotes a non-activated housekeeping microglial state. In vitro and in vivo rodent studies have shown that CX3CL1 reduces microglial activation markers and the secretion of anti-neurogenic pro-inflammatory cytokines such as TNF-α, Il-1β, and IL-6 [114,115]. The beneficial effects of CX3CL1 have also been demonstrated in the hippocampus, where impairment of CX3CR1 signaling was shown to disrupt rodent AHN, spatial and fear-associated memory, and motor learning; driven at least partly through increased secretion of IL-1β by activated microglia [112,116,117]. In addition to anti-inflammatory effects, CX3CL1 signaling in microglia may produce positive effects on synaptic maintenance and function; depletion of either microglia or CX3CR1 was shown to reduce dendritic spine density and pruning in the mouse olfactory bulb [118]. Interestingly, CX3CL1 also imparts a survival advantage to in vitro cultured human NPCs in the absence of growth factors [119].

Damaged or apoptotic cells can express a range of “eat-me” endogenous cell-death-related products. Recognition of such apoptotic cells by glia occurs through DAMP-PRR interactions that activate targeted pro-inflammatory and phagocytic responses. “Eat-me” signals include the expression of complement proteins C1q and C3 [120] or the externalization of phosphatidylserine, which then acts as a DAMP that signals through the PRR Triggering Receptor Expressed on Myeloid cells 2 (TREM2), mainly present on microglia [121]. Various other PRRs can bind danger signals, including the TLRs, RIG-I-like receptors (RLR), nucleotide-binding oligomerization domain (NOD)-leucine rich repeats containing receptors (NLR), and C-type lectin receptors (CLR), retinoic acid-inducible gene 1 (RIG1)-like receptors (RLR), and the receptor for advanced glycation end-products (RAGE) on immune cells triggering responses aimed at restoring tissue and organism homeostasis [122,123,124]. In the neurogenic niche, PRRs are expressed on NPCs, neurons, and glia and provide critical communication pathways from apoptotic cells as well as after injury and infection [125,126]. As such, it is important to recognize that PRRs not only modulate glial activity but also that of neurons themselves. For instance, activation of neuronal TLR3, TLR7, and TLR8 is important for cell differentiation and dendritic and axonal morphology [127,128]. In embryonic murine NPCs, TLR3 activation inhibits proliferation, while TLR4 inhibition reduces in vitro human NPC/NSC proliferation [129,130]. Thus PRR signaling is implicated in the modulation of adult neurogenesis across the physiological-pathophysiological spectrum.

In addition to the absence of “eat-me” signals, cells can also express “don’t eat me” signals. One of the best characterized is the membrane glycoprotein CD200 (formerly OX-2) and its receptor CD200R [131,132]. In the rodent and human nervous systems, CD200 is expressed by neurons while its cognate receptor is embedded in various immune cells, including microglia and some astrocytes [132,133,134,135]. Activation of CD200-CD200R signaling attenuates microglial activation and reduces the secretion of the anti-neurogenic pro-inflammatory cytokines IL-1β and TNF-α [134,136,137,138].

#### 3.1.2. The Nuclear Receptor TLX

An important but somewhat unexpected player in microglial modulation is TLX (Nuclear Receptor Subfamily 2 Group E Member 1; NR2E1). TLX is one of the key drivers of AHN. In the brain, TLX is expressed in and regulates the function of NSCs/NPCs in the SGZ and SVZ, maintaining them in an undifferentiated proliferative state [139,140,141]. This is mediated through the regulation of multiple proteins and transcription factors, such as repression of the tumor suppressor gene pten (restrains stem cell proliferation) and the cyclin-dependent kinase inhibitor p21 (restrains cell proliferation; maintains quiescence), while activating Achaete-scute homolog 1 (ASCL1 or MASH1; cell cycle regulator and promoter of proliferation) and Wnt/β-catenin signaling (promoting proliferation and self-renewal) [142,143,144,145]. Thus, hippocampal TLX expression correlates with cell proliferation and increased neurogenesis [146]. As such, TLX is critical to normal embryonic brain development and AHN alike [147].

Besides its cell-intrinsic regulatory functions, TLX also appears to regulate the local inflammatory microenvironment. In mice, TLX attenuates the expression of pro-inflammatory genes and particularly the expression of the anti-proliferative cytokine IL-1β [148,149,150]. In essence, while TLX is not a recognized point of direct cell-to-cell contact, its expression appears to act as a no-danger signal and thus restrain pro-inflammatory glial responses through unclear mechanisms. This relationship makes sense because maintaining glia in a non-activated optimally-phagocytic state assists with the removal of surplus cells while minimizing off-target damage. Mouse models have shown that microglia involved in this sort of housekeeping phagocytosis remain in exactly this unchallenged non-activated state [42]. Interestingly, part of the TLX-microglial cross-talk may involve modulation of CX3CR1-CX3CL1 signaling. In one recent study, CX3CR1 knockout in mice was shown to be associated with reduced TLX transcription [151]. However, it is not yet known if TLX expression itself affects CX3CL1 secretion. If true, this would explain how a nuclear receptor can affect microglial behavior through what is essentially paracrine signaling.

One of the limitations in our ability to properly characterize TLX has been a lack of known endogenous ligands. Several natural and synthetic molecules have been identified as potential TLX ligands, demonstrating both agonist and inverse agonist effects in vitro [152,153]. Recently, the monounsaturated omega-9 fatty acid oleic acid was identified as a TLX endogenous ligand, synthesized by NSCs/NPCs to trigger their cell cycle and promote neurogenic progeny [154]. This is consistent with the general ability of nuclear receptors, such as steroids and phospholipids, to bind lipophilic molecules such as steroids and phospholipids [155,156]. The identification of an endogenous ligand validates TLX as a therapeutic target and should facilitate further mechanistic insights and the development of therapeutic compounds that would promote AHN and potentially suppress microglial neuroinflammation.

### 3.2. Pathophysiological Danger Signaling

#### 3.2.1. Grow or Die: The Complexity of PRR Danger Signaling

Neurological injury, aging, neurodegeneration, and other pathological states are associated with anti-neurogenic and pro-inflammatory shifts in cellular function and the microenvironment. In addition to the expression of previously mentioned “find me” signals, cell stress can result in the release of alarmins such as High Mobility Box Group 1 (HMGB1), S100B, and heat shock proteins (HSP). These are expressed in NSCs, can promote apoptosis in hippocampal neurons, and when released they prime microglia toward a pro-inflammatory state [157,158,159,160]. The excessive release of the alarmin ATP during epileptic seizures impairs phagocytosis and reduces cell clearance in brain sections obtained from rodents as well as human patients undergoing temporal lobectomy for mesial temporal lobe epilepsy [44]. It was posited that excessive ATP signaling disrupted the finely-tuned chemotactic gradients that normally help microglia locate apoptotic cells. Following some neurological insults, alarmin levels may even be utilized as clinical biomarkers; for instance, plasma levels of HMGB1 and S100B have been correlated with poor outcomes in patients with aneurysmal subarachnoid hemorrhage and traumatic brain injury [161,162,163,164].

When acting on neurons themselves, alarmins can produce pro-neurogenic effects. In the mouse brain, HMGB1 is critical for embryonic neurogenesis and to stimulate NPC proliferation and differentiation through the PRR Receptor for Advanced Glycation End products (RAGE), which then activates NF-κB, a highly complex transcription factor involved in inflammation and growth [165,166]. Similarly, intraperitoneal infusion of S100B enhances AHN in a mouse model of traumatic brain injury, by stimulating proliferation, migration, and neuronal differentiation [167]; HSP70 was similarly shown to increase hippocampal cell proliferation and differentiation and improve learning and memory in the object recognition test in mice [168].

In turn, the transmembrane PRR glycoproteins TLR2 and TLR4 have similarly complex and contextual effects. Activation of these receptors by gram-positive or gram-negative bacterial products, respectively, in innate immune cells such as microglia, results in NF-κB activation and the release of pro-inflammatory cytokines. Correspondingly, brain injury in animal models activates microglial inflammatory responses via TLR [169,170,171] while signaling from damaged neurons via HGMB1-TLR4 plays a role in neurite degeneration and cognitive impairment in 5xFAD mice that model Alzheimer’s Disease (AD) amyloid pathology [172]. However, TLR expression and activation in adult NPCs and neurons have been associated with neuroplasticity. Using mouse knockout models, it was demonstrated that TLR2-deficient NPCs exhibit impaired differentiation and, instead, are more likely to switch toward gliogenesis [126]. These effects were shown to act via MyD88-dependent activation of NF-κB. Conversely, in the same study, TLR4-deficiency was associated with increased NPC proliferation and neuronal differentiation.

Other PRRs, such as the RLRs RIG-I and MDA5 (melanoma differentiation-associated protein 5), have also been found on NSCs as part of a defense system against infection by viruses such as Japanese encephalitis and Zika, the latter of which has been associated with dramatically impaired embryonic neurogenesis that culminates in microcephaly [173]. By infecting NSCs/NPCs, neurons, and glia, Zika exerts many effects that include increased pro-inflammatory cytokine release, found in amniotic fluid of affected mothers [174]. Although we are not aware of research into the potential role of RIG-I-like PRRs in AHN, their reported pro-apoptotic effects in viral infections and glioblastoma multiforme suggest that they could play physiological and pathological roles in cell maintenance and development in other contexts, including AHN [175]. Other PRRs such as TLR9 and TLR4 are important; microglial TLR9 detects damage-associated neuronal self-DNA and attenuates seizure-related AHN, while astrocytic TLR4 activation has been reported to enhance excitatory synaptogenesis resulting in greater seizure activity [176,177]. It is worth noting, however, that there is some controversy as to whether astrocytes express TLR4 [178,179].

On the whole, the outcome of PRR responses is dependent on where they are expressed and in what context they are activated. Much more work remains to be conducted to fully delineate these mechanisms, especially in the context of AHN.

#### 3.2.2. No Danger Signaling: CD200 in Pathology

As previously described, CD200 is a “no danger” signal that may provide a target for the treatment of neuropathology. In a model of AD, utilizing transgenic mice carrying the Swedish familial AD mutant of human amyloid precursor protein APP695, CD200 suppresses microglial inflammation, enhances microglial phagocytosis, attenuates the loss of NPC proliferation and differentiation, and promotes dendritic density, thus preventing the loss of spatial learning and memory abilities [180,181]. Activation of CD200 via a CD200 fusion protein attenuates age-related hippocampal microglial activation and long-term potentiation (LTP) deficits in rats [182]. The importance of CD200 in human neuropathology is supported by reports of reduced CD200-CD200R expression in the cerebrospinal fluid and brains of individuals affected by AD and in Multiple Sclerosis lesions, suggesting that this mechanism is part of the pathophysiology of human neurodegenerative chronic neuroinflammation [135,183,184].

### 3.3. Glial Inflammatory Responses to Danger Signals

Once activated by DAMPs, PAMPs, or an existing inflammatory environment, astrocytes and microglia respond by releasing pro-inflammatory and anti-neurogenic cytokines such as IL-1β and TNF-α. Cytokines are secreted glycoproteins that comprise an essential cell-cell communication pathway during immune responses to injury or infection, physiological surveillance, and tissue maintenance. Their advantageous and detrimental roles in AHN and relevant neuropsychiatric disorders are well recognized. It is important to note that a key characteristic of cytokines and cytokine networks is their functional complexity in physiology and pathophysiology. This is due to their pleiotropic (multiple and differential effects) and redundant (shared effects) natures, as well as contextual and temporal effects (what is beneficial in one instance may be harmful in another). In addition, glia can phagocytose damaged and dying cells. However, this activity can result in off-target damage to healthy non-apoptotic neurons; one such demonstrated mechanism involves the release of lactadherin (MFG-E8) by neurons, a molecule that bridges phagocytes with target cells [185].

#### 3.3.1. The Interleukin-1 Family

The seven-member IL-1 family includes four pro-inflammatory cytokines that appear to play a particularly important role in modulating AHN: IL-1β, IL-1ra, IL-18, and IL-33 [186]. Most secreted IL-1β in the brain originates from microglia; this often occurs in response to IL-1β itself, although microglia do not themselves express the IL-1 receptor (IL-1R1) [187,188]. IL-1R1, a member of the TLR family, is expressed on blood-brain-barrier endothelial cells, choroidal cells, and at low levels in astrocytes [187]. Binding of IL-1β to IL-1R1 results in myeloid differentiation primary response protein 88 (MyD88)-dependent and MyD88-independent pathway activation, including induction of c-Jun N-terminal kinase (JNK), p38 MAPK, and NFκB, thus modulating proliferation, cell survival, apoptosis, and immune responses [189,190]. While IL-1β has myriad immunological and non-immunological effects on the brain, including induction of neurotrophic factors that may protect or harm neurons depending on various physiological and temporal factors, the effects on adult neurogenesis appear to be uniformly negative [191]. The importance of IL-1β in AHN was recognized over a decade ago when it was found that its receptor IL-1R1 is expressed on neurons and NPCs in the hippocampus and dentate gyrus, that IL-β expression has anti-proliferative effects, and that hippocampal IL-1β mediates the anti-neurogenic effects in animal models of stress [192,193,194,195,196,197,198,199].

Correspondingly, the IL-1 receptor antagonist (IL-1ra), an inhibitor of IL-1, attenuates the effects of IL-1β and interestingly, has been linked to neuropsychiatric disorders such as schizophrenia and depression, whose etiologies appear to involve aberrant AHN [200,201,202,203]. In vitro treatment of rat hippocampal NPCs with IL-1ra increases TLX receptor expression and attenuates IL-1β-induced reductions in TLX expression and NPC proliferation [148,196]. Similarly, intracerebroventricular injection of IL-1ra into rat brains blunted the anti-proliferative effects of subsequently administered IL-1β and foot-shock stress [193]. In a West Nile Virus model of neuronal dysfunction, murine intraperitoneal administration of the FDA-approved drug Anakinra, a recombinant form of IL-1ra, attenuated the loss of spatial learning [204]. Similarly, intracerebroventricular injection of recombinant IL-1ra into rat brains attenuated fear conditioning deficits associated with social isolation [205]. Unfortunately, neither of these studies evaluated any direct effects on neurogenesis.

IL-1β may also exert its effects on AHN indirectly. For instance, as previously described, there appears to be a bidirectional relationship between IL-β and the TLX receptor expression that modulates AHN; secreted IL-1β should decrease TLX expression [148,149,150,206,207]. IL-1β also alters the microglial functional state toward a pro-inflammatory phenotype, thus potentially resulting in the release of additional anti-neurogenic immunomodulatory products [187].

IL-18 is activated in response to PAMP and DAMP signaling, binds to the TLR receptor IL-18r, and activates NFκB signaling through MyD88-dependent mechanisms, eventually resulting in the release of pro-inflammatory products [208,209]. IL-18 is produced in astrocytes and activated microglia, while expression of its receptor has been somewhat inconsistently reported on microglia, astrocytes, oligodendrocytes, and neurons [91,210,211,212,213,214]. Constitutive expression of IL-18r expression is generally reported to be low and upregulated following tissue damage (e.g., in astrocytes during astrogliosis) [91,214,215]. Research into the role of IL-18 in AHN is limited and somewhat conflicting, and its in vivo effects are not yet well defined. On the one hand, IL-18 appears to reduce NPC differentiation and survival in vitro but it is not known if this effect is reproduced when IL-18 is released by activated microglia in vivo [216]. On the other hand, IL-18 correlates with an increase in new neurons following exercise, potentially through pro-angiogenic effects [217,218].

IL-33 is a relatively recently identified nuclear cytokine that is released as a pro-inflammatory alarmin following cellular injury [219]. In the brain, IL-33 is expressed in endothelial cells, oligodendrocytes, astrocytes, and neurons [220,221,222]. The IL-33 receptor, IL1RL1 (formerly ST2), exists as a soluble antagonist receptor (IL1RL1a; sST2) and as a membrane-bound TLR (IL1RL1b; ST2) that, when activated by extracellular IL-33, results in MyD88-dependent NFκB signaling [223]. In the brain, IL1RL1 is expressed in astrocytes, microglia, and possibly in neurons [221,224,225]. As an alarmin and NFκB activator, extracellular IL-33 mainly serves as a pro-inflammatory stimulus resulting in microglial activation, proliferation, and enhanced phagocytosis [221]. Enhanced production of IL-1β, TNF-α, and several chemokines suggests that IL-33 should be inherently unfavorable to neurogenesis [221]. However, recent research has revealed that neuronal and astrocytic IL-33 is critical to microglial synaptic maintenance and plasticity [224,225]. Reduced IL-33-IL1RL1 communication due to genetic knockout or aging is correlated with a reduction in newborn neurons following environmental enrichment in mice [225], while lentivirus-induced overexpression of IL-33 in aged mice rescues the phenotype [225]. It remains imperative to define the source, temporal, dose, and other contextual parameters of this cytokine in the context of hippocampal stem cell proliferation, differentiation, maturation, and integration to gauge a more precise role of this cytokine in AHN regulation.

#### 3.3.2. Tumor Necrosis Factor (TNF)

TNF-α is an immensely complex, pleiotropic, classically pro-inflammatory cytokine that exists in soluble (sTNF) and transmembrane (tmTNF) forms thus mediating both direct and indirect cell-cell communication [226]. tmTNF is a precursor form that is cleaved by the metalloprotease TNF-α converting enzyme (TACE) into sTNF that ultimately binds to its receptors TNFR-I (TNFR60; CD120a) and TNFR-II (TNFR80; CD120b) [226,227,228]. Both TNF forms activate TNFR-I and TNFR-II, although TNF activity appears to be primarily mediated through TNFR-II [229]. Activation of TNFR-I activates multiple downstream signal pathways resulting in its multiple functional effects including the promotion of inflammation, cell survival, proliferation [NFκB and mitogen-activated protein kinases (MAPK)], and apoptosis [caspase-8 and poly(ADP-ribose) polymerase (PARP)] [230,231]. TNFR-II activation is typically associated with pro-survival regenerative signaling through activation of signaling pathways such as NFκB [232]. However, the functional effects of TNFR-II are complex and may depend on cell type as demonstrated in a murine experimental autoimmune encephalomyelitis (EAE) model of multiple sclerosis, where microglial TNFR-II was protective while its expression on infiltrating myeloid cells was pathogenic [233]. In addition, TNF receptors can be shed from the cell surface by proteolytic cleavage into soluble forms, sTNFR-I and sTNFR-II, that both suppress excessive TNF-α activity [234].

In AHN, the effect of TNF-α appears to at least partially depend on its target receptor. NPCs appear to express both TNFR-I and TNFR-II [235,236]. In genetic knockout mouse models of the two receptors, TNFR-I suppressed NPC proliferation while its knockout increased in vitro and in vivo NPC proliferation. Conversely, TNFR-II promoted NPC proliferation [237,238]. These findings were corroborated in a murine sciatic nerve injury pain model known to induce microglial activation. This injury resulted in increased hippocampal TNF-α and reduced expression of TNFR-II, thereby reducing the formation of newborn neurons; these effects were absent in TNFR-I knockout mice [239,240].

## 4. Key Research Questions

We have discussed some of the key glial communication and response pathways that modulate AHN by exploiting danger signaling as a central theme. We have shown that many of the presented molecules and cell functions serve to both pamper (boost) and dampen adult neurogenesis. We would like to end this review by highlighting some of our most pressing questions and providing a needs list for the field.

### 4.1. Danger Signaling and Responses in AHN

We have discussed many of the key molecules, pathways, and cell-cell interactions that participate in the pathogen- and danger-associated pathways and the current understanding of how these factors affect neurogenesis. However, much remains unknown about the AHN in physiology and pathophysiology. For example, how do glia coordinate their unique and overlapping functions to preserve NPCs and newborn neurons? During aging or in pro-inflammatory states, can we permanently modify anti-neurogenic glial cell phenotypes and thus help rescue neurogenesis, while maintaining normal beneficial glial function? The net benefit or harm of functions such as inflammation and phagocytosis are highly contextual; while critical to pathogen elimination and the development of efficient, functional neuronal circuits and the removal of toxic proteins such as tau and amyloid β, attenuation of pro-inflammatory cytokine release and phagocytosis would be an important component of a therapeutic approach to treat aging-related cognitive decline by increasing AHN [241]. A comprehensive understanding and fine control of these mechanisms will be critical to the development of therapies that aim to increase AHN (e.g., for age-related cognitive decline). For instance, to generate and integrate a meaningful amount of new neurons into the aging brain, we may need to transiently attenuate inflammation and phagocytosis to maximize cell survival. This is particularly critical because relatively few NSC/NPC are present in the old brain, and thus maximizing cell proliferation is likely to be an important factor. Answers to these questions are likely to arise from the combined efforts of biological inquiry and sophisticated modeling of the AHN cascade and microenvironment [242].

### 4.2. Cell Identity and Relationships

While much progress has been made, the lack of identified unique live-cell markers still presents a significant roadblock to basic and translational research [243,244]. The transition from NSC, to NPC, to neuroblast, and to mature neuron involves a spectrum of change that is associated with gradual shifts in morphology, the transcriptome, proteome, and metabolome, including changes in subcellular organelles such as mitochondria and autophagosome, while the newborn cells in moving from the SGZ into the granule cell layer terminal location. Unfortunately, these transitions are associated with the expression of common cell surface markers among closely related cells, hampering unambiguous identification, particularly when downstream viability is studied.

Specific labeling of cell types is particularly critical to the development of a clearer understanding of how microenvironmental variables and cell-to-cell communication modulate cell proliferation, maturation, and survival during the generation of new functional neurons. The availability of specific cell surface proteins would also provide potential binding and interaction sites for the investigation of cell-to-cell and cell-to-microenvironment communications. Approaches such as fluorescent labeling and proximity ligation assays could then be used to investigate specific protein–protein interactions in vivo or in environments that closely replicate the in vivo state. Finally, specific cell markers may be amenable to translation to cell labeling for in vivo use (e.g., positron emission tomography ligands), which could revolutionize our approaches in animal and human studies and even present as clinical biomarkers.

### 4.3. Human AHN

Granted that the recently awakened controversy about the veracity of human AHN has been resolved in favor of its existence, there remain several unanswered questions about in vivo human neurogenesis [1]. 

The most critical of these concerns relates to the clinical implications of modulating AHN. First, will therapeutic augmentation of AHN translate to improvements in cognitive function? Animal studies suggest that enhancements of cell proliferation and differentiation are associated with behavioral improvements, but we still do not know what happens if AHN is enhanced for long periods of time, months or even years. Until this is tried in humans, we will not know if and how much these functional effects can be recapitulated. If direct modulation is attempted for psychiatric disorders where aberrant AHN and neuroplasticity have occurred, will neurogenesis impart clinically-beneficial effects? Again, animal studies showing that AHN is a critical component of the anti-depressive response following electroconvulsive shock or anti-depressants suggests that this should be the case [245], but while human postmortem studies indicate diminished AHN in depressed patients, we do not know whether antidepressants will exert the same mechanism of action as they do in rodent models.

The second question concerns the potential adverse effects of stimulating neurogenesis. The proliferative role of TLX is recognized in the field of oncology as an adverse prognostic marker and potential treatment target for prostate cancer and glioblastoma [246,247,248]. Thus, could pharmacological stimulation of neurogenesis, for example through TLX modulation, result in uncontrolled cell proliferation? Current data suggest that TLX expression, while a risk factor in established cancers, is on its own insufficient to generate pathological cell proliferation. However, any therapies targeting TLX to increase NSC/NPC proliferation will need to ensure that neurogenesis remains confined within tight physiological limits and that no tumor formation elsewhere happens [248].

We also need to better understand how physical and psychological life events perturb AHN in humans. For instance, how do life events such as stress and infections modulate glial-NSC/NPC-neuronal cell interactions and signaling that promote and suppress neurogenesis in the developed brain? What role, if any, do sex differences play? Significant sex-based differences have been reported in the neuroimmune drivers of pain and may also be relevant in other areas including neurodegeneration and neurogenesis [249]. Finally, many clinical multi-symptom disorders such as fibromyalgia, chronic fatigue syndrome, and Gulf War Illness — and as we are all witnessing SARS-CoV-2 — present with concomitant cognitive abnormalities and appear to be associated with neuroimmune dysfunction [250,251,252,253]. Whether neurogenesis is impacted by such aberrant neuroimmune function and thereby plays a role in the pathogenesis or treatment of these disorders remains to be examined. These questions again underlie the critical need for biomarkers that can be used to test the function of the neurogenic niche in humans, in vivo.

In this article, we have reviewed the interactions between cells of the hippocampal niche by utilizing the immunological concept of “danger signaling”. We find this to be a helpful foundation to integrate the myriad pro- and anti-neurogenic signals that occur during the AHN cascade. Ultimately, we hope that a deep understanding of this signaling and its complex effects on AHN will allow us to halt and repair aging- and neurodegenerative-related cognitive decline.

## Data Availability

Not applicable.

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
