# Peer review of "Glial PAMPering and DAMPening of Adult Hippocampal Neurogenesis"

_brainsci, 2021, doi:10.3390/brainsci11101299_

Round 1
Reviewer 1 Report
In the review manuscript entitled ‘Glial PAPering and DAMPering of Adult Hippocampal Neurogenesis’ by Parkitny and Maletic-Savatic, the authors survey the existing knowledge on glia-mediated regulation of adult hippocampal neurogenesis (AHN) and critically discuss missing evidence and pending questions. This is a particularly well written manuscript that attempts to fill a literature gap on the complex and multicellular regulatory correlates of AHN with particular emphasis on inflammatory and other toxic stimuli.
As the authors discuss, recent studies firmly demonstrated the existence of AHN in human brain and its possible functional correlation with brain disease, such as AD. However, knowledge on the mechanistic underpinnings of these observations is almost solely obtained from rodent studies. The role of glial cell populations at the neurogenic niche is of particular significance herein.
Some remarks are listed below:
- While particularly relevant to the scope of this review, the authors do not always discuss putative links of the described cell types/molecular signaling to disease states. A reference to the relevance of a particular pathway to disease (as that provided for CD200-CD200R to AD) would be highly beneficial.
- For some of the discussed signaling pathways, there is evidence on NPC-mediated expression (of some of the signaling molecules) and regulation, via which e.g. immunomodulatory signals are exerted from RGLs/NPCs to glial cells. The authors should add some discussion on this when possible.
- When discussing marker specificity or the lack thereof, some mention of recent scRNAseq studies (e.g. Hochgerner et al on mouse dentate) should be included and used to draw conclusions.
- Along these lines, GFAP is not only an astrocytic marker, but is also highly expressed by RGLs in the dentate niche.
- The authors should be clear when implying that there is evidence of human AHN also in the SVZ, which is not a given at this point.
- The term ‘ANP’ for neuronal precursors is not a standard one. Unless, the authors would argue against it, I would propose to replace it with a more commonly used one.
- Line 268: The authors mistakenly refer to the APPPS1 mouse model as ‘APPS’ mice.
- The sentences in lines 118 and 139/142 would profit from some rephrasing.
9. There is a full stop missing in line 54.
Reviewer 2 Report
In this review paper entitled "Glial PAMPering and DAMPening of Adult Hippocampal Neurogenesis", Parkitny and Maletic-Savatic describe general aspects of adult hippocampal neurogenesis and the pathogen- and danger-associated pathways.
The manuscript is well written, but the discussion of the literature is superficial and contains established concepts and general descriptions. To bring something new and attract the interest from readers, the authors should better discuss the literature, and especially elaborate more on the PAMP and DAMP aspect.
Besides that, there are some points of minor concern:
- The authors should always indicate whether the findings described derive from animal (rats/mice), in vitro, or clinical/translational studies.
- The sentences in line 45 and in lines 55-56 are very similar and could be merged together.
- A verb is missing in line 140.
- In line 232 the authors describe "hippocampal apoptosis". Which cell types?
- In line 236, the authors could change "improve object reconition" to "improve learning & memory in the object recognition test".
- In line 451, the subtopic is denominated "Neurogenesis in vivo", but perhaps the authors wanted to indicate translational/clinical studies?
